# Impaired Intracellular Ca^2+^ Dynamics, M-Band and Sarcomere Fragility in Skeletal Muscles of Obscurin KO Mice

**DOI:** 10.3390/ijms23031319

**Published:** 2022-01-24

**Authors:** Enrico Pierantozzi, Péter Szentesi, Cecilia Paolini, Beatrix Dienes, János Fodor, Tamás Oláh, Barbara Colombini, Dilson E. Rassier, Egidio Maria Rubino, Stephan Lange, Daniela Rossi, László Csernoch, Maria Angela Bagni, Carlo Reggiani, Vincenzo Sorrentino

**Affiliations:** 1Department of Molecular and Developmental Medicine, Molecular Medicine Section, University of Siena, 53100 Siena, Italy; pierantozzi@unisi.it (E.P.); egidiomaria.rubino@student.unisi.it (E.M.R.); daniela.rossi@unisi.it (D.R.); 2Department of Physiology, Faculty of Medicine, University of Debrecen, H-4002 Debrecen, Hungary; szentesi.peter@med.unideb.hu (P.S.); dienes.beatrix@med.unideb.hu (B.D.); fodor.janos@med.unideb.hu (J.F.); olahtamas@gmail.com (T.O.); csl@edu.unideb.hu (L.C.); 3Department of Neuroscience, Imaging and Clinical Sciences, University Gabriele d’ Annunzio of Chieti, 66100 Chieti, Italy; cecilia.paolini@unich.it; 4Department of Experimental and Clinical Medicine, University of Florence, 50134 Florence, Italy; barbara.colombini@unifi.it (B.C.); mangela.bagni@unifi.it (M.A.B.); 5Department of Kinesiology and Physical Education, McGill University, Montreal, QC H2W 1S4, Canada; dilson.rassier@mcgill.ca; 6Biomedical Research Facility 2, School of Medicine, University of California, La Jolla, CA 92093, USA; slange@health.ucsd.edu; 7Department of Molecular and Clinical Medicine, Institute of Medicine, University of Gothenburg, 413 45 Gothenburg, Sweden; 8Department of Biomedical Science, University of Padova, 35121 Padova, Italy; carlo.reggiani@unipd.it; 9Science and Research Center Koper, Institute for Kinesiology Research, 6000 Koper, Slovenia

**Keywords:** obscurin, skeletal muscle, calcium dynamics, muscle fiber damage, kinetics of contraction, sarcoplasmic reticulum, exercise

## Abstract

Obscurin is a giant sarcomeric protein expressed in striated muscles known to establish several interactions with other proteins of the sarcomere, but also with proteins of the sarcoplasmic reticulum and costameres. Here, we report experiments aiming to better understand the contribution of obscurin to skeletal muscle fibers, starting with a detailed characterization of the diaphragm muscle function, which we previously reported to be the most affected muscle in obscurin (*Obscn*) KO mice. Twitch and tetanus tension were not significantly different in the diaphragm of WT and *Obscn* KO mice, while the time to peak (TTP) and half relaxation time (HRT) were prolonged. Differences in force-frequency and force-velocity relationships and an enhanced fatigability are observed in an *Obscn* KO diaphragm with respect to WT controls. Voltage clamp experiments show that a sarcoplasmic reticulum’s Ca^2+^ release and SERCA reuptake rates were decreased in muscle fibers from *Obscn* KO mice, suggesting that an impairment in intracellular Ca^2+^ dynamics could explain the observed differences in the TTP and HRT in the diaphragm. In partial contrast with previous observations, *Obscn* KO mice show a normal exercise tolerance, but fiber damage, the altered sarcomere ultrastructure and M-band disarray are still observed after intense exercise.

## 1. Introduction

The *OBSCN* gene encodes a giant sarcomeric protein, obscurin, originally identified as a titin-interacting protein [1]. In striated muscles, two high-molecular-weight obscurin variants, known as obscurin A (~720 kDa) and obscurin B (970–870 kDa), are present. Both proteins are characterized by a modular architecture consisting of tandem immunoglobulin (Ig) and fibronectin III (Fn3)-like domains, followed by some signaling motifs, including a calmodulin IQ-binding motif, a src homology3 (SH3) domain, a RhoGEF and a pleckstrin homology (PH) domain. These two variants differ at the C-terminal region, with obscurin A, the predominant isoform in skeletal muscles, being characterized by a non-modular sequence containing a binding site for ankyrin proteins [2,3,4]. In obscurin B, the non-modular sequence present in the C-terminal region of obscurin A is replaced by a longer amino acid stretch containing two C-terminal serine/threonine kinase-like domains, further expanding the potential signaling functions of obscurin proteins [3,5,6,7,8]. In addition to the two giant obscurin A and B, additional obscurin isoforms of a smaller molecular weight have also been detected in non-muscle tissues, although the physiological role of these non-muscular obscurin isoforms is not yet known [9].

In skeletal muscle fibers, the obscurin localization pattern reveals a predominant distribution at the M-band, and a minor one at the Z-disk. At the Z-disk region, obscurin interacts with titin, with a likely preference for the smaller titin isoform novex-3 (∼700 kDa), forming a complex that may regulate myofibrillar signaling pathways involved in stress-initiated sarcomeric restructuring [10,11,12]. At the M-band of the sarcomere, obscurin directly interacts with titin, myomesin and myosin binding protein-C generating a protein matrix which is responsible for the maintenance of the overall sarcomere integrity during repeated cycles of contraction [1,5,13,14,15]. Accordingly, in both obscurin knockout (*Obscn* KO) mice and Unc-89 (i.e., the homolog of giant obscurin in *Drosophila* sarcomere) knockdown fruit flies, a relevant number of fibers presented severe alterations of the M-band [16,17,18]. The ankyrin binding domains present in the obscurin A allow the interaction with the small muscle-specific isoform of Ankyrin 1 (sAnk1.5; ≈20 kDa), which is localized on the sarcoplasmic reticulum membrane, preferentially in correspondence to the M-band, but also to the Z-disk. The physical interaction between these two proteins tethers the sarcoplasmic reticulum around the myofibrils; thus, ensuring alignment and a stable connection of the sarcoplasmic reticulum during repetitive cycles of sarcomere contraction and relaxation [4,14,19,20,21,22]. Indeed, the initial characterization of *Obscn* KO mice revealed a significant shortening of the longitudinal sarcoplasmic reticulum, paralleled by decreased sAnk1.5 protein levels [23]. Interestingly, in a mirror-like image, skeletal muscle fibers of sAnk1.5 KO mice also presented a strong reduction in the sarcoplasmic reticulum volume [24]. The ankyrin binding domains in obscurin also interact with the muscle-specific Ank2.2 isoform, encoded by the *ANK2* gene [25,26,27]. In skeletal muscle, this interaction appears to contribute to the proper localization and organization of dystrophin at costameres, likely by providing a physical connection between the sarcomere, the microtubular cytoskeleton and the sub-sarcolemmal protein network, which would contribute to the maintenance of the overall fiber integrity during contraction [27,28,29]. In fact, in muscle fibers from *Obscn* KO mice, Ank2.2 loses its localization in correspondence of the M-band, the sub-sarcolemmal microtubular network is altered, and dystrophin is no longer targeted to the costameres, leading to sarcolemmal fragility [27]. A few variants in the *OBSCN* gene have been identified in patients with cardiomyopathy and distal myopathy [30,31,32,33]. However, a direct association between the presence of these genetic variants and human diseases remains to be further validated [34].

While the structural role of obscurin in supporting sarcomere assembly and maintenance as well as providing a connection with multiple cellular components is well defined, its contribution to force generation and contraction has been, to date, only partially elucidated. Both in vivo and ex vivo experiments on gastrocnemius or EDL muscles revealed no difference both in the force generation and in activation and recovery kinetics in *Obscn* KO muscle fibers when compared with controls. This was apparently in strong contrast with the limited running capability of *Obscn* KO mice when required to perform intense running protocols on a treadmill [3,23,27,35].

To understand the role of obscurin in generating muscle force and contractions, we perform a detailed analysis of the contractile properties of the diaphragm muscle from *Obscn* KO mice. Indeed, previous results indicated that the diaphragm muscle is the more severely affected muscle by obscurin ablation [3,27]. Given the evidence that in skeletal muscle fibers from *Obscn* KO mice the sarcoplasmic reticulum volume reduces [23], we verify whether Ca^2+^ signaling is affected by the deletion of obscurin [36]. Finally, we also re-evaluate the running capability of the *Obscn* KO mice and again investigate the impact of deletion on the structural integrity of the M-band and of diaphragm muscle fibers.

## 2. Materials and Methods

### 2.1. Animals

The generation of *Obscn* KO mice has been described in Lange et al., 2009 [23]. The mouse colony was maintained by intercrossing heterozygous mice, and the genotype of the progeny was determined by polymerase chain reaction amplification, as previously described [23]. Experiments were performed using control (WT) and *Obscn* KO male mice weighing 30–40 g (7–10 months old). The animals were kept under standard laboratory conditions (12 h light/dark cycle, room temperature 22–25 °C) with free access to tap water and pelleted mouse chow in their plastic cages, which had mesh covers. All the reported procedures, including mice euthanasia, were approved by the Animal Care Committee of the University of Siena (OBPA) and authorized by the Italian Ministry of Health, in compliance with directive 2010/63/EU of the European Parliament and the Council of 22 September 2010 about welfare of animals used for scientific purposes.

### 2.2. Voluntary Running and Treadmill Exercise

Voluntary wheel running was used to test exercise capacity. Mice were housed in individual cages equipped with running wheels (Campden Instruments Ltd., Loughborough, UK), placed in a dark room with 12 h periods of dark and light, and free access to food and water. Wheels were interfaced to a computer and revolutions were recorded at 20 min intervals, continuously for 14 days. The overall distance run was calculated for each mouse and then averaged by groups (*n* = 4 mice per genotype). Furthermore, running ability of WT and *Obscn* KO mice was tested with a treadmill (LE8709MTS; Panlab, S.L.U., Barcelona, Spain) equipped with an electrified grid, using a heavy uphill running protocol [27] and a running exercise performed to exhaustion [3].

### 2.3. EBD Uptake Assay

Evans blue dye (EBD) (E2129, Sigma-Aldrich^®^, St Louis, MO, USA) was injected intraperitoneally (1 mg/0.1 mL *per* 10 g of body weight), as previously described [16]. Mice were sacrificed after 24 h, EDL muscle was removed and snap frozen using Optimum Cutting Temperature compound (OCT, Sakura Finetek, Torrance, CA, USA) in nitrogen-cooled isopentane at −165 °C. Cross-sections (20 μm) were prepared to evaluate the extent of EBD uptake. Positive fibers were imaged with a Plan-Neofluar 20X, 0.50 NA objective using the LSM 510 META confocal microscope and the LSM image browser (Carl Zeiss, Jena, Germany).

### 2.4. Contractile Performance of Ex Vivo Diaphragm Bundles

Mice were sacrificed by quick cervical dislocation to minimize their suffering. The diaphragm muscle still attached to lower ribs was quickly removed and placed in Tyrode solution ((mM): NaCl, 121; KCl, 5; CaCl_2_, 1.8; MgCl_2_, 0.5; NaH_2_PO_4_, 0.4; NaHCO_3_, 24; glucose, 5.5; EDTA, 0.1) and bubbled with 5% CO_2_–95% O_2_, which gave a pH of 7.4. Fetal calf serum (0.2%) was routinely added to the solution.

Small bundles of up to fifteen intact fibers were dissected from the diaphragm as described previously [37]. The use of small bundles of fibers (thereafter called bundles, for simplicity) offered ideal conditions for mechanical measurements. The dissection was performed manually under a stereo-microscope with a fine pair of scissors and needles, taking care to avoid stretching and to obtain preparations clean of debris from dead fibers. Small aluminum T-shaped clips were fixed to tendons as close as possible to the bundles’ ends. The bundles were transferred to a temperature-controlled experimental chamber (801C/1900, Aurora Scientific, Aurora, ON, Canada) and mounted on an inverted microscope (Axiovert 40CFL, ZEISS, Germany). The clips attached the bundles horizontally between the lever arms of a capacitance force transducer (405A, Aurora Scientific) and a motor (322C, Aurora Scientific). Bundles were super-fused continuously by means of a peristaltic pump with the oxygenated Tyrode solution. The experiments were performed at 27–28 °C in Tyrode solution. Bipolar stimuli (1 ms duration and 1.5 times threshold strength) were applied across the bundles by means of two platinum-plate electrodes mounted parallel to the bundle via a high power bipolar stimulator (701C, Aurora Scientific). After a test of viability, tetanic stimulation (500 ms duration; 125 Hz) was applied to set the bundles length at the optimal sarcomere length (about 2.70 µm) to obtain the maximal tetanic force. Tetani were evoked at an interval of 1 min for a period of equilibration of about 10 min, in which the maximum tetanic force (P_0_) remained stable. If P_0_ decreased by 15%, the bundles were discarded. Length, largest and smallest diameters and resting sarcomere length of the bundles were measured using the microscope fitted with 20× eyepieces and a 5× or 40× dry objective. Cross-sectional area of the bundles was calculated as a*b*π/4, where ‘a’ and ‘b’ are the average values of the width and the vertical diameters, measured at 2–3 different points. For all measurements of muscle force and length, the output of force transducer and length controller were acquired in real time in the PC by using an integrated board (NI6221, National Instruments, Austin, TX, USA). The experimental data were displayed and successively analyzed by a commercial software (600A, Aurora Scientific).

Twitch contractions elicited by a single stimulus, 1 ms duration, were analyzed to determine peak isometric twitch tension (Pt, expressed in mN/mm^2^), time-to-peak (TTP) and half relaxation time (HRT). Maximal isometric tetanic tension (P_0_, expressed in mN/mm^2^) was determined on the average of three contractions at 125 Hz stimulation frequency, 500 ms duration. The force-frequency relation was obtained with isometric contractions at 15, 30, 40, 50, 75, 100, 125 and 150 Hz, with trains of 500 ms duration, at 1 min interval. Force at each stimulation rate was expressed as a fraction of maximal isometric tetanic tension (P_0_). 

### 2.5. Fatigue and Recovery in Diaphragm

Muscle fatigue was defined as the decline in isometric force development in repeated stimulations. The rate of fatigue development was determined by monitoring the decrease in isometric force production at 40 Hz that is considered the diaphragm physiological frequency of contraction. Unfused tetanic contractions (500 ms duration, 40 Hz stimulation rate) were elicited every 1 s (duty ratio 1:2) for 500 consecutive tetani. After the fatigue protocol, the force recovery was followed by evoking tetanic contractions (500 ms, 40 Hz) every 5 min for a period of 30 min.

### 2.6. Force-Velocity Relationship in Diaphragm

Force-velocity (P-V) data were obtained using the controlled release method. The bundles stimulated at high frequency to develop maximum tetanic tension were allowed to shorten at preset velocity (V) and the associated steady isotonic tension (P_i_) was measured. A quick release was applied before the constant velocity release to reduce the time needed to attain a steady isotonic force [38]. To assess muscle function after the P-V protocol, two maximal tetanic contractions were evoked and compared with the P_0_ before the P-V data collection. The P-V parameters were obtained by fitting the Hill’s equation [39] with the software OriginPro 2015 (OriginLab Corp., Northampton, MA, USA)
V = [(1 + *a*/P_0_) *b*/(Pi/P_0_)] − *b*(1)
where V is the shortening velocity, P_i_ is the isotonic tension and *a*/P_0_ and *b* are constants. The maximum shortening velocity (V_max_) was calculated by extrapolating the fitted curve to zero tension. Power output was determined by the product of each pair of force and shortening velocity values.

### 2.7. Transmission Electron Microscopy

#### 2.7.1. Muscle Preparation

Wild-type and *Obscn* KO mice at 6 months of age (n = 6 per genotype) were euthanized immediately after the completion of the last intense running session of the uphill running protocol. Diaphragm was excised and processed as previously described [3,40]. Briefly, diaphragm muscles were fixed in situ at room temperature in 3.5% glutaraldehyde in 0.1 M sodium cacodylate buffer, pH 7.2, then dissected and kept in fixative until further use. Small strips of fixed diaphragms were isolated and post-fixed in 2% OsO_4_ in the same sodium cacodylate buffer for 2 h and block-stained in aqueous saturated uranyl acetate. After dehydration, specimens were embedded in an epoxy resin (EPON 812). Ultrathin sections (40 nm) were cut in a Leica Ultracut R microtome (Leica Microsystem, Vienna, Austria) using a Diatome diamond knife (Diatome A.G., Bienne, Switzerland). After being stained in lead citrate, sections were examined with a Morgagni Series 268D electron microscope (FEI, Brno, Czech Republic) equipped with a Megaview III digital camera. Ultrastructural alterations and organization of the M-band with respect to the sarcomere and ultrastructural damages of the fibers were analyzed.

#### 2.7.2. Quantitative Analyses of EM Specimens

Determination of M-band damages, or of the number of damaged fibers in WT and *Obscn* KO mice, was performed on longitudinally oriented EM micrographs. Micrographs, all at the same magnification (18,000×) with no overlapping regions, were analyzed and individual intact fibers were identified as delimited by sarcolemma surrounding the sarcoplasm. Each fiber was visually scored for: (1) ultrastructural alterations and organization of the M-band with respect to the sarcomere; (2) the presence of unstructured cores and/or contracture cores, as described in Paolini et al., 2015 [41]. Quantification of the number of fibers with alterations was presented as a percentage of all analyzed fibers.

### 2.8. Ca^2+^ Measurements on Single Flexor Digitorum Brevis (FDB) Muscle Fibers

All Ca^2+^ concentration measurements were carried out on single intact FDB fibers of WT and *Obscn* KO mice. Single muscle fibers from FDB were enzymatically dissociated in a minimal essential medium containing 0.2% Type I collagenase (Sigma, St Louis, MO, USA) at 37 °C for 35 min [42]. To obtain single intact fibers, FDB muscles were triturated gently in normal Tyrode’s solution (1.8 mM CaCl_2_). Fibers were then placed in culture dishes and stored at 4 °C in refrigerator until use.

#### 2.8.1. Intracellular Ca^2+^ Concentration Measurements

The resting intracellular Ca^2+^ concentration ([Ca^2+^]_i_) was measured using Fura-2 fluorescent calcium indicator as described previously [42]. Briefly, enzymatically isolated single FDB fibers were mounted on a laminin-coated glass cover slip and loaded with 5 µM Fura-2 acetoxymethyl ester (AM) for 1 h. Fibers were then kept in normal Tyrode’s solution for half an hour at room temperature. Measurements were carried out on an inverted fluorescence microscope (Nikon Diaphot, Nikon, Tokyo, Japan). A microcomputer-controlled dual-wavelength monochromator alternated the excitation wavelength with 50 Hz between 340 and 380 nm (DeltaScan, Photon Technology International, New Brunswick, NJ, USA), whereas a photomultiplier measured the emission signal at 510 nm with a 10 Hz acquisition rate at room temperature. [Ca^2+^]_i_ was assessed as the ratio of measured fluorescence intensities (F340/F380). Resting [Ca^2+^]_i_ was calculated from averaging the ratio in normal Tyrode’s.

#### 2.8.2. Voltage Clamp and Ca^2+^ Transient Analysis

The experimental design was similar to the one previously described [43]. Briefly, isolated fibers were voltage-clamped (Axoclamp 2B, Axon Instruments, Foster City, CA, USA) and imaged using a confocal microscope (ZEISS LSM 5 Live) in external solution (in mM: 140 TEA-CH_3_SO_3_, 1 CaCl_2_, 3.5 MgCl_2_, 10 Hepes, 1.4-AminopyridineP, 0.5 CdCl_2_, 0.3 LaCl_3_, 0.001 TTX, and 0.05 BTS, pH 7.2, osmolality 320 mOsm). Fibers were dialyzed with rhod-2 containing internal solution (in mM: 110 N-methylglucamine, 110 L-glutamic acid, 10 EGTA, 10 Tris, 10 glucose, 5 NaATP, 5 phosphocreatine Tris, 0.1 rhod-2, 3.56 CaCl_2_, and 7.4 mM MgCl_2_, pH 7.2, osmolality 320 mOsm). All experiments were conducted at room temperature (20–22 °C) and the resting holding potential was set at −80 mV. The stimulation protocol contained a train of 100 ms long depolarizations from −60 to +30 mV with +10 mV increment in every second. The pipette resistance varied in the range of 1–2 MΩ. Analog compensation was used to correct the linear capacitive currents.

Ca^2+^ transients were analyzed by fitting a removal model that calculated release flux. This took into account the evolution of [Ca^2+^]_i_(t) in a single compartment, including quantitatively specified processes of removal, such as the maximal rate of the SERCA pump (PV_max_) [44,45]. The experiments were performed in the presence of calcium buffer (10 mM EGTA); thus, the endogenous buffers were considered almost negligible in the removal process. The Boltzmann function was used to describe the voltage dependence of the activation of calcium Ca^2+^ release:Δ[Ca^2+^]_i,max_/(1 + exp(−(V_m_ − V_50_)/k))(2)
where V_50_ is the midpoint voltage and the reciprocal of k is the limiting logarithmic of the slope. Line-scan images were analyzed by a custom-made program using the following parameters: K_d_ rhod-2 = 1.58 μM and k_ON_ = 0.7·10^8^ 1/M/s. From the release flux (the flux exiting through the release channels), the net flux leaving the sarcoplasmic reticulum could be derived by subtraction of the pump removal flux. The integral of the net flux provided the sarcoplasmic reticulum content releasable by depolarization or otherwise termed the amount of Ca^2+^ released (Δ[Ca^2+^]_tot_).

### 2.9. Immunofluorescence Labeling of EDL Muscle Fibers

Mechanically isolated Extensor Digitorum Longus (EDL) fibers from both WT and KO mice were fixed and permeabilized as previously described [27]. A monoclonal antibody against α-actinin (clone EA-53, Sigma-Aldrich, St Louis, MO, USA) was used to identify the Z-disks, and a polyclonal antibody against dystrophin (EB-9024, Thermo Fisher Scientific, Waltham, MA, USA) was used to immunolocalize dystrophin. Cy3-conjugated anti-rabbit or Alexa Fluor488 anti-mouse secondary antibodies (Thermo Scientific) were used for immunofluorescence detection. Fibers were imaged using the LSM510 META confocal microscope with a Plan-Neofluar 20× and 63× 0.50 NA objective, and acquired with the LSM acquisition software (Carl Zeiss, Jena, Germany).

### 2.10. SDS Page and Immunoblotting

Total protein lysates from diaphragm and FDB muscles were prepared and quantified as previously described [24]. SDS page and immunoblotting were performed as in [24], with minor modifications. Briefly, 30 or 50 micrograms of total lysates were loaded either on 12% TGX stain-free FastCast acrylamide electrophoresis gel (Bio-Rad, Hercules, CA, USA) or on 4–12% gradient electrophoresis gel (Invitrogen, Carlsbad, CA, USA). Separated proteins were transferred onto nitrocellulose membrane. Total protein load was revealed by either Ponceau Red staining and UV detection of the stain-free membrane. Filters were blocked with 5% non-fat dried milk in TRIS-buffered saline and TWEEN 20 for 1 h at room temperature. Primary antibody incubation was performed overnight at 4 °C with anti-pan-SERCA and anti-sAnk1.5 [4] antibodies. Following extensive washing of the membranes, secondary anti-rabbit- and anti-mouse antibodies conjugated to horseradish peroxidase (Cell Signaling Technology, Danvers, MA, USA) were incubated 1 h at room temperature. Immunoreactive bands were revealed using the ECL Star (Euroclone, Pero, Italy), and acquired by the Chemidoc Imaging System (Bio-Rad, Hercules, CA, USA). Quantification of the intensities of immunoreactive bands was performed by Image Lab software (Bio-Rad, Hercules, CA, USA), using optical densities of total protein bands pattern as normalizer. 

### 2.11. Statistical Analysis

Data were expressed as mean ± standard error of the mean (SEM) or ± standard deviation (SD). Comparisons between experimental groups (WT vs. *Obscn* KO) for each variable investigated were carried out with unpaired Student’s *t*-test. Forces developed by the WT and *Obscn* KO bundles of diaphragm at different frequencies of stimulation or during the fatigue protocol were compared by two-way ANOVA, followed by Bonferroni’s post hoc test. F-test was used to test the significance and a *p* value of less than 0.05 was considered statistically significant. All statistical analyses were performed with Prism (GraphPad Software, San Diego, CA, USA).

## 3. Results

### 3.1. Contractile Performance of Diaphragm Ex Vivo

Previous experimental data pointed to the diaphragm being more sensitive to the deletion of obscurin than hind limb muscles [16,27,35]. To further characterize the effects of obscurin deletion on muscle contractile performance, we chose ex vivo bundles of diaphragm intact fibers as the experimental model. Isometric tension developed in the twitch and tetanus (at 40 Hz and 125 Hz stimulation) was slightly, but not significantly, lower in *Obscn* KO than in WT mice (Table 1 and Figure 1A,B). Twitch time parameters (i.e., time to peak (TTP) and half relaxation time (HRT)) in diaphragm bundles are reported in Figure 1C,D and Table 2. The TTP and HRT were significantly prolonged in *Obscn* KO diaphragm bundles compared to WT controls, indicating that the lack of obscurin resulted in overall slower twitch kinetics in this muscle, possibly related to a slower calcium removal by the SERCA (see below).

A change in the force-frequency relationship could be expected in accordance with prolonged twitch kinetics. Actually, this was confirmed by recording the isometric force at increasing stimulation rates, from 15 to 150 Hz with trains of 500 ms duration, at 1 min intervals, to avoid fatigue. These measurements were repeated before and after a fatigue protocol (see below) in eight WT and in nine *Obscn* KO bundles. The results are shown in Figure 2. Before fatigue, the force-frequency relationship was rather similar between the two experimental groups, although, compared to WT, bundles from *Obscn* KO mice displayed a slight upward shift in the range from 20 to 80 Hz. Such differences became statistically significant when the force-frequency relationship was obtained at the end of the recovery period, following the completion of the fatigue protocol. Of note, the force generated by both WT and *Obscn* KO bundles was reduced at any stimulation frequency after fatigue and recovery (see below).

Since the prolonged twitch kinetics were suggestive of an impaired sarcoplasmic reticulum function in *Obscn* KO muscle fibers compared to WT, we asked whether the observed difference could be enhanced by a fatiguing protocol. It is widely accepted that fatigue alters the sarcoplasmic reticulum function, increasing calcium leakage while reducing evoked calcium release and calcium re-uptake [46]. Fatigue was induced by reducing the time interval between tetani at 40 Hz, 500 ms duration, to the fatiguing interval of 1s (duty ratio 1:2), and the decay of the developed tension was recorded during 500 consecutive tetani. Since the force developed by WT and *Obscn* KO bundles was different at the beginning of the protocol (see Table 1 and Figure 1A), all force values were normalized relatively to the maximal force developed at 40 Hz (P_40_) before fatigue. The decline in the developed force was significantly greater in *Obscn* KO diaphragm bundles compared to WT (Figure 3, Fatigue panel). At the end of the fatigue protocol, the remaining force was 0.50 ± 0.06 P_40_ in WT bundles and 0.28 ± 0.03 P_40_ in *Obscn* KO bundles (*p* < 0.05). The analysis of variance confirmed that fatigue from the 20th to the 500th tetanus was significantly higher (*p* < 0.05) in *Obscn* KO bundles compared to WT (Figure 3, Fatigue panel).

Although fatigue led to different levels of force in *Obscn* KO and WT diaphragm bundles, both experimental groups displayed comparable recovery patterns when stimulated at 40 Hz, 500 ms duration, with long rest intervals of 5 min. After 1 min of recovery, the tetanic force generated by WT bundles was still greater than that of *Obscn* KO bundles, 0.81 ± 0.06 P_40_ and 0.63 ± 0.05 P_40_ (*p* < 0.05). After five minutes following the end of fatigue, all bundles reached the maximum force recovery value, and then the developed tension remained virtually constant. At the end of the recovery period, the tetanic force was 0.87 ± 0.02 P_40_ and 0.90 ± 0.03 P_40_ in WT and *Obscn* KO bundles, respectively (Figure 3, Recovery panel).

To characterize the function of the myofibrillar motors, we determined the force-velocity relationship during maximal isometric contractions in WT and *Obscn* KO diaphragm bundles. Pairs of force and velocity data were obtained by measuring the isotonic force (P_i_) during a ramp release at different velocities (V). Data points were fitted to Equation (1) [39] and are shown in Figure 4. The maximum shortening velocity (V_max_) obtained by extrapolation from the fitting was 3.44 ± 0.41 l_0_/s for WT (*n* = 7) and 2.84 ± 0.38 for *Obscn* KO (*n* = 6), but this difference was not statistically significant (*p* = 0.311). The ratio a/P_0_, which expresses the curvature of the relationship, was higher in WT (0.36 ± 0.03) than in *Obscn* KO bundles (0.27 ± 0.03) (*p* < 0.05), the difference being clearly visible in Figure 4. The values of optimal force and shortening velocity, i.e., the force and the velocity at which peak power is reached, was 1.17 l_0_/s at 0.34 P_i_/P_0_ for WT and 0.90 l_0_/s at 0.32 P_i_/P_0_ for *Obscn* KO diaphragm bundles, respectively.

### 3.2. Altered Intracellular Calcium Homeostasis

The initial characterization of the *Obscn* KO mice [23] showed that the absence of obscurin affected the sarcoplasmic reticulum structure and resulted in reduced sAnk1.5 protein levels. Notably, the deletion of sAnk1.5, in addition to inducing a reduction in the sarcoplasmic reticulum volume [24], also resulted in an altered sarcoplasmic reticulum Ca^2+^ release [47]. Based on that, the altered contractile kinetics observed in muscles from *Obscn* KO mice may be explained with an impairment of intracellular calcium dynamics. Accordingly, we measured the resting intracellular Ca^2+^ concentration ([Ca^2+^]_i,rest_) in Fura-2-loaded Flexor Digitorum Brevis (FDB) fibers excised from WT and *Obscn* KO mice. [Ca^2+^]_i,rest_ was significantly increased in KO (*n* = 26) compared to WT fibers (*n* = 34) (82.7 ± 2.8 nM vs. 70.4 ± 0.8, respectively; *p* < 0.001). We also measured the changes in [Ca^2+^]_i_ elicited under whole-cell voltage clamp by progressively increasing membrane depolarization between −60 mV and +30 mV. As shown in Figure 5A,B,D the average change in the amplitude of Ca^2+^ transients at +30 mV depolarization was significantly lower in *Obscn* KO than in WT mice (Table 3).

As reported in Figure 5C, the voltage dependence of the transients was not affected by obscurin deletion. The midpoint voltage (V_50_) and the slope of the Boltzmann curves (k) was similar in both genotypes (Table 3). To determine the SERCA pump activity, the falling phase of the maximal calcium transients was fitted with a calcium removal model (see the Section 2). The maximal speed of the SERCA pump was found to be significantly (*p* < 0.001) lower in *Obscn* KO (549.6 ± 29.8 µM/s) than in WT FDB fibers (748.1 ± 36.5 µM/s) (Figure 5E).

These latter results prompted us to evaluate SERCA protein levels in *Obscn* KO muscles (Figure 5F,G). An immunoblot analysis of the FDB total protein lysates revealed that SERCA expression levels were significantly reduced by about 60% in *Obscn* KO mice compared to WT. Interestingly, a similar robust reduction in SERCA expression was observed in diaphragm muscles. Notably, the decrease in sAnk1.5 protein levels was found to be more pronounced in KO diaphragm than in KO FDB, compared to WT muscles (Figure 5F,G).

### 3.3. Normal Running Ability of Obscn KO Mice

Next, we wanted to verify the ability of *Obscn* KO mice to perform spontaneous physical activity when housed in cages equipped with a running wheel for fourteen days. Unexpectedly, *Obscn* KO mice displayed voluntary running performance almost identical to that of WT animals (data not shown). This result prompted us to test the running ability of *Obscn* KO mice also using treadmill-based running protocols, since the results on voluntary running were in contrast with our own previous observations of reduced running performance. Actually, we consistently observed a reduced running performance when *Obscn* KO mice were tested by different protocols, including intense uphill running and exhaustion protocols, on a treadmill apparatus [16,27]. However, as shown in Figure 6, when *Obscn* KO mice were also tested using the intense running exercise on a treadmill, no difference was observed in their running performance with respect to WT mice.

### 3.4. Altered H-Zone and M-Band in Obscn KO Mice after Exercise

We previously reported that muscle fibers of *Obscn* KO mice presented evidence of a reduced dystrophin localization at costameres as a consequence of the loss of interaction with Ank2.2 [27,35]. The reduced localization of dystrophin at costameres was associated with an exacerbated sarcolemmal fragility of *Obscn* KO muscles, as revealed by the Evans Blue Dye (EBD) uptake analysis in muscle fibers of KO mice that completed an intense uphill running protocol and in ex vivo electrically stimulated fibers. Exercise also resulted in an increased number of contracture knots and the loss of organization and/or disassembly of the M-band in the diaphragm muscle [16,27]. 

As shown in Figure 7F–K, we observed a dystrophin reduction at costameres in muscle fibers of *Obscn* KO mice. Accordingly, we evaluated the sarcolemmal fragility of *Obscn* KO skeletal muscle fibers through the detection of EBD-positive fibers, following an intraperitoneal injection of EBD immediately before the last running session of the intense uphill running protocol. This analysis confirmed that a higher percentage of EBD-positive fibers was present in *Obscn* KO EDL muscle (203/1216 EBD^+^ fibers; 16.69%) compared to WT controls (29/758 EBD^+^ fibers; 3.82%).

The morphological analysis also confirmed that intense physical exercise resulted in evident M-band alterations and Z-disk streaming, or severe fiber damages (i.e., contractures, unstructured cores, sarcomere disarrays) in *Obscn* KO diaphragm fibers (Figure 7A–E). Following the execution of an intense uphill running protocol, a TEM analysis of the M-band in sarcomeres of *Obscn* KO fibers showed structural alterations as well as the loss of sharpness in 34% of the observed fibers (*n* = 177 fibers from six mice, Figure 7B) compared to WT fibers, where the M-band was correctly placed at the center of sarcomeres and aligned with that of adjacent myofibrils (*n* = 188 WT fibers from six mice, Figure 7A). In addition to alterations in the M-band, in 19% of *Obscn* KO diaphragms fibers, severe structural alterations in the sarcomere organization, such as unstructured cored and sarcomere disarray (Figure 7C), areas of extreme sarcomere shortening (contracture cores) were observed (Figure 7D). Therefore, these data indicated that even if the *Obscn* KO mice appeared to have rescued tolerance to perform intense running activities, the deletion of obscurin significantly affected the overall ability of myofibrils to maintain their integrity when exposed to an intense work load.

## 4. Discussion

Here, we reported experiments aimed at determining the effects of obscurin deletion on the contractile properties and Ca^2+^ kinetics in ex vivo muscle fibers, along with additional experiments where we re-evaluated the running ability of *Obscn* KO mice and the effects of intense exercise on the stability of the contractile apparatus in diaphragm muscle.

In previous work, we reported that *Obscn* KO mice displayed a reduced sarcoplasmic reticulum volume, decreased localization of dystrophin at costameres, reduced exercise tolerance and sarcolemma and M-band fragility following intense exercise [16,23,27,35]. In those experiments, we observed that the diaphragm, but not the hind limb muscles, was the most-affected muscle when *Obscn* KO mice were requested to perform intense running protocols [16]. Here, we reported that the analysis of force development revealed that neither twitch tension nor tetanic force were altered in the diaphragm muscle from *Obscn* KO. This was not surprising, considering that the skeletal muscle fiber size, length, type and muscle mass, the major parameters on which the overall ex vivo skeletal muscle tension development relies [48], were not altered in *Obscn* KO with respect to WT mice [16,23,27,35]. In contrast, we found that kinetics parameters of muscle contraction, TTP and HRT, which strongly depend on intracellular Ca^2+^ dynamics, were prolonged in KO mice. Accordingly, the force-frequency relationship was slightly shifted to the left. Interestingly, the difference in the force-frequency relationship between the *Obscn* KO and WT diaphragm muscles was even more pronounced at the end of the fatigue protocol, which represented a substantial challenge for the sarcoplasmic reticulum function. This further supports the view that obscurin (and its binding partner sAnk1.5) are relevant for the correct sarcoplasmic reticulum function. The analysis of sarcoplasmic reticulum Ca^2+^ release in FDB muscle fibers showed that the absence of obscurin reduced the amount of Ca^2+^ released from the sarcoplasmic reticulum without affecting the ability of the fiber to properly integrate the depolarizing stimulus. This implied that the observed impairment of Ca^2+^ release appeared to reflect intrinsic sarcoplasmic reticulum defects of *Obscn* KO fibers, in agreement with the reported reduction in the longitudinal sarcoplasmic reticulum [23]. This evidence suggested that the structural involvement of obscurin in preserving sarcoplasmic reticulum architecture was also relevant to sarcoplasmic reticulum Ca^2+^ handling, as previously proposed in a study on the invertebrate obscurin homolog unc-89 [49]. Similarly to *Obscn* KO muscle fibers, skeletal muscle fibers lacking sAnk1.5, namely, the obscurin interactor on the sarcoplasmic reticulum side, were characterized by a significant reduction in the longitudinal sarcoplasmic reticulum volume [24], and displayed an impaired sarcoplasmic reticulum Ca^2+^ release that almost completely mirrored the one we observed in *Obscn* KO mice [47], further indicating that the maintenance of sarcoplasmic reticulum architecture was a pivotal prerequisite for E–C coupling to occur properly. A further indication of the relevance of sarcoplasmic reticulum alterations came from the reduced tolerance to repeated stimulations. It is widely accepted that the contractile force depression, which is a direct measurement of muscle fatigue, is determined by an impaired calcium release from the sarcoplasmic reticulum [50]. Thus, the greater fatigability of *Obscn* KO muscles could be explained as a consequence of the negative effect of the reduced sarcoplasmic reticulum volume on Ca^2+^ handling.

The analysis of Ca^2+^ handling in FBD fibers also revealed that a resting cytosolic Ca^2+^ concentration was significantly increased in *Obscn* KO FDB fibers compared to WT fibers. By using previously validated mathematical models [44,45], we inferred that the maximal activity of the SERCA pump was significantly slowed down in *Obscn* KO fibers. This may be explained by the reduction in SERCA protein levels observed in *Obscn* KO FDB fibers. In agreement with previous data [23], we noted that sAnk1.5 protein levels were decreased in the *Obscn* KO muscles analyzed, including FDB. This may represent an additional mechanism in regulating SERCA activity, since the direct binding of sAnk1.5 to SERCA was reported to regulate Ca^2+^ transport by SERCA [51]. On the other hand, whether the sAnk1.5/SERCA interaction might result either in potentiating or in lowering the pump activity still needs to be fully elucidated, considering that sAnk1.5 appears also to counteract the effect of sarcolipin, an inhibitor of SERCA activity [52]. Alterations in Ca^2+^ cycling have also been observed in the cardiomyocytes of knock-in mice carrying the obscurin R4344Q mutation identified in a patient with cardiomyopathy [30,53]. In this case, alterations in Ca^2+^ handling were suggested to result from phospholamban being sequestered by binding to obscurin, although this mechanism has been debated [34,53]. Nevertheless, irrespective of whether additional mechanisms, other than the more established role of obscurin in maintaining and stabilizing the sarcoplasmic reticulum around myofilaments, may contribute to the impairment of Ca^2+^ homeostasis in *Obscn* KO mice, the reported data clearly indicated a requirement of obscurin in maintaining proper Ca^2+^ handling in skeletal muscle fibers [36,50,54,55].

The last results reported here allowed the re-evaluation of the running ability of the *Obscn* KO mice that, at variance with what we previously reported, displayed an exercise tolerance not significantly different from that of WT mice. The only way for us to explain this unexpected finding is that, over time, *Obscn* KO mice had overcome the initially observed inability to perform intense exercises. A trend in improving the running ability of these mice could already be observed by comparing the reported results of the running test observed in our reports after a few years. Indeed, one could note that, in the more recent report, an improvement in the exercise tolerance of *Obscn* KO mice could be observed with respect to the initial report. We cannot exclude that the previous results could have been originated by a non-voluntary repetitive inbreeding of the mice in the experiments reported [16,27]. Indeed, inbred mouse colonies can be affected by additional variables, including spontaneous variations, which are difficult to identify [56,57,58]. On this note, it is worth underlining that the experiments reported in this manuscript were obtained by carefully crossing heterozygous mice and that experiments were also repeated with the progeny of new *Obscn* KO founders, kindly provided again by Dr. Ju Chen at the University of California, San Diego. Moreover, we cannot rule out the possibility that additional events, including changes in the diet which may affect mice microbiota, may have contributed over the years to modify the mouse phenotype [59]. All in all, we are rather inclined to explain the changes in the running ability of the *Obscn* KO mice as some sort of adaptation. On the other hand, the levels of the expression of the obscurin-like1 gene, Obsl1, a paralog of obscurin, were shown to be increased in the *Obscn* KO mice [35]. Faced with this unexpected recovery of the running ability of *Obscn* KO mice, we repeated dystrophin staining of the plasma membrane, EBD staining and morphological evaluation of muscles, following intense exercise on a treadmill. As a result, we found that, even if *Obscn* KO mice had reacquired a normal tolerance to exercise such as that of WT mice, they still presented a plasma membrane and M-band fragility, following intense exercise on a treadmill.

In summary these data indicated that force parameters in *Obscn* KO mice were not significantly altered, but twitch kinetic parameters were prolonged. The altered contraction kinetics could be explained by the changes of the Ca^2+^ transients observed, possibly related to the reduced sarcoplasmic reticulum volume and a slower Ca^2+^ removal by the SERCA pumps. In addition, even if over time the *Obscn* KO mice appeared to have reclaimed the tolerance for performing intense running activities, we confirm that obscurin is required to maintain the integrity of diaphragm muscle fibers when mice are exposed to an intense work load, and that the M-band represents the most sensitive sarcomere region in *Obscn* KO mice.

## Figures and Tables

**Figure 1 ijms-23-01319-f001:**
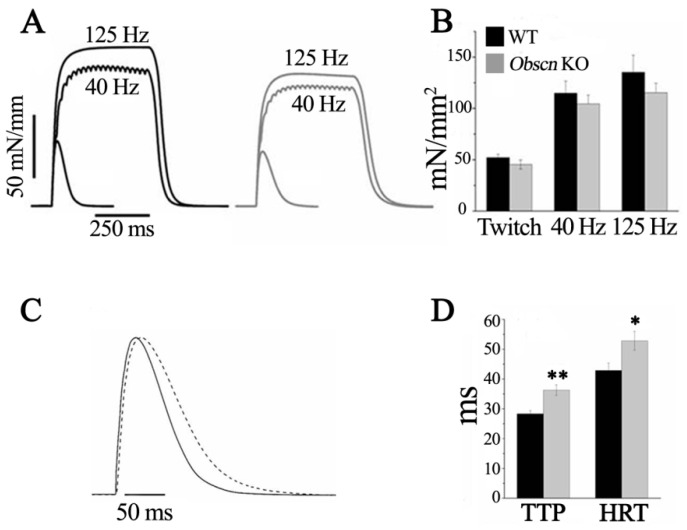
Isometric contractions in fiber bundles of diaphragm of WT and *Obscn* KO mice. (**A**) Representative records of twitch and tetanic contractions of WT (black) and *Obscn* KO (grey) fiber bundles isolated from diaphragm. (**B**) Bars report the isometric peak tension values (mean ± SEM). (**C**) Enlarged records of twitch contractions in WT (continuous line) and *Obscn* KO (dashed line) diaphragm bundles (tension values were normalized for the maximum twitch tension in muscles from each genotype in order to compare the different time courses). (**D**) Bars report the mean values ± SEM of time to peak (TTP) and half relaxation time (HRT). * *p* < 0.05 and ** *p* <0.01 compared to WT.

**Figure 2 ijms-23-01319-f002:**
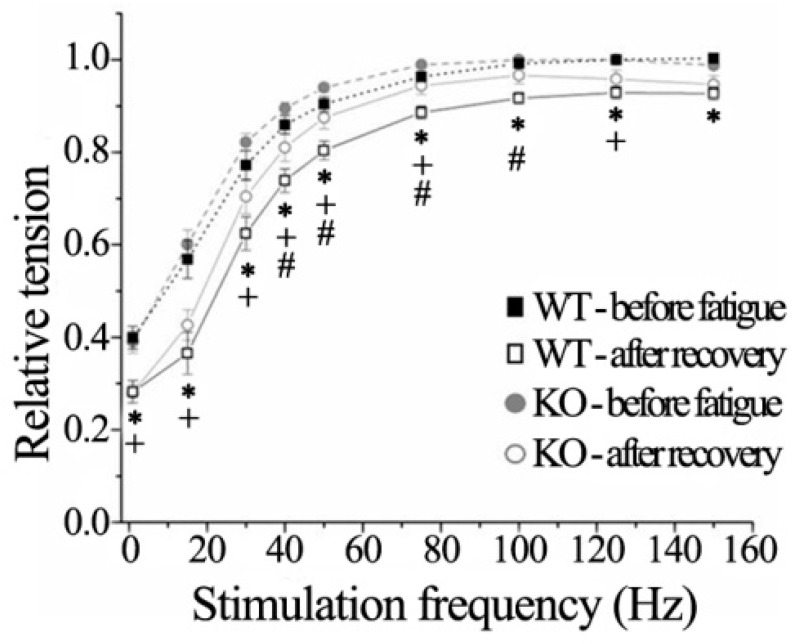
Force-frequency relationship of WT and *Obscn* KO diaphragm fiber bundles. Force-frequency relationship was determined in WT (*n* = 8) and *Obscn* KO (*n* = 9) bundles before fatigue and after the completion of the fatigue–recovery protocol. All force values were normalized against maximal tension (P_0_) before fatigue at 125 Hz. Values are reported as mean ± SEM. Statistically significant difference (*p* < 0.05) was as follows: *—WT bundles before vs. after fatigue; +—KO bundles before vs. after fatigue; #—KO vs. WT bundles after fatigue.

**Figure 3 ijms-23-01319-f003:**
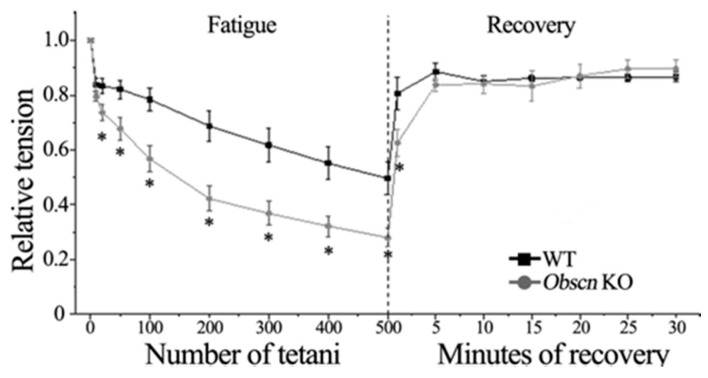
Fatigue and recovery in WT and *Obscn* KO diaphragm bundles. Average time course of tension during fatigue and recovery of WT (black squares; *n* = 8) and KO (grey circles; *n* = 9) bundles. Tension values are expressed relative to the maximal force developed at 40 Hz (P_40_) before fatigue as mean ± SEM. * = *p* < 0.05 *Obscn* KO compared with the corresponding values of WT bundles.

**Figure 4 ijms-23-01319-f004:**
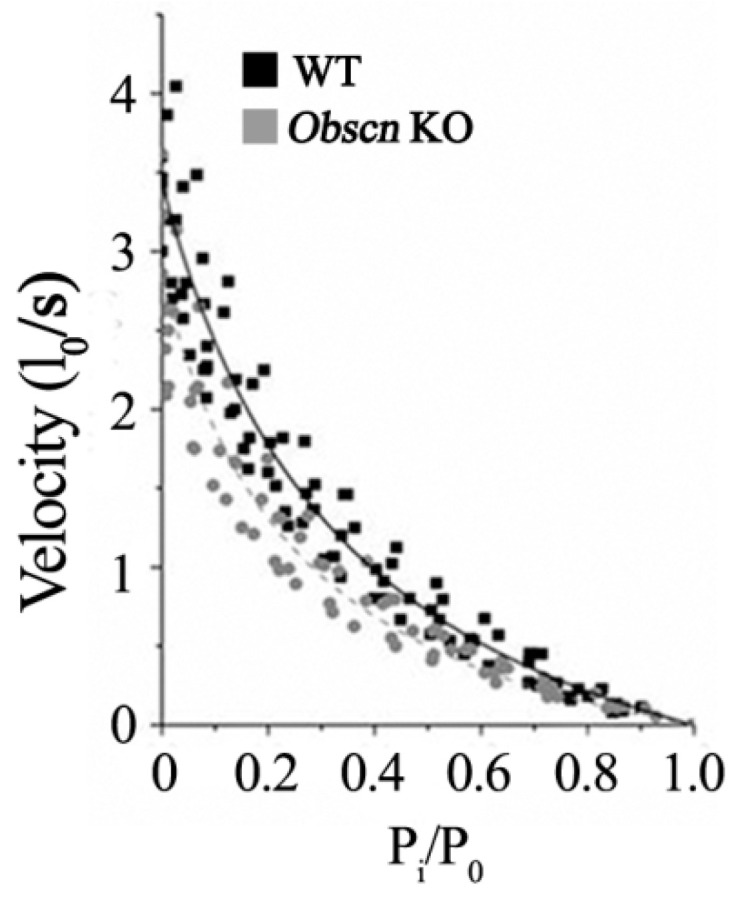
Force-velocity relationships in WT and *Obscn* KO diaphragm bundles. Force-velocity relationship in WT (black squares; *n* = 7) and KO (grey circles; *n* = 6) diaphragm muscle bundles. Continuous (WT) and dashed (KO) curves were obtained from the fitting of Hill’s equation to the experimental data.

**Figure 5 ijms-23-01319-f005:**
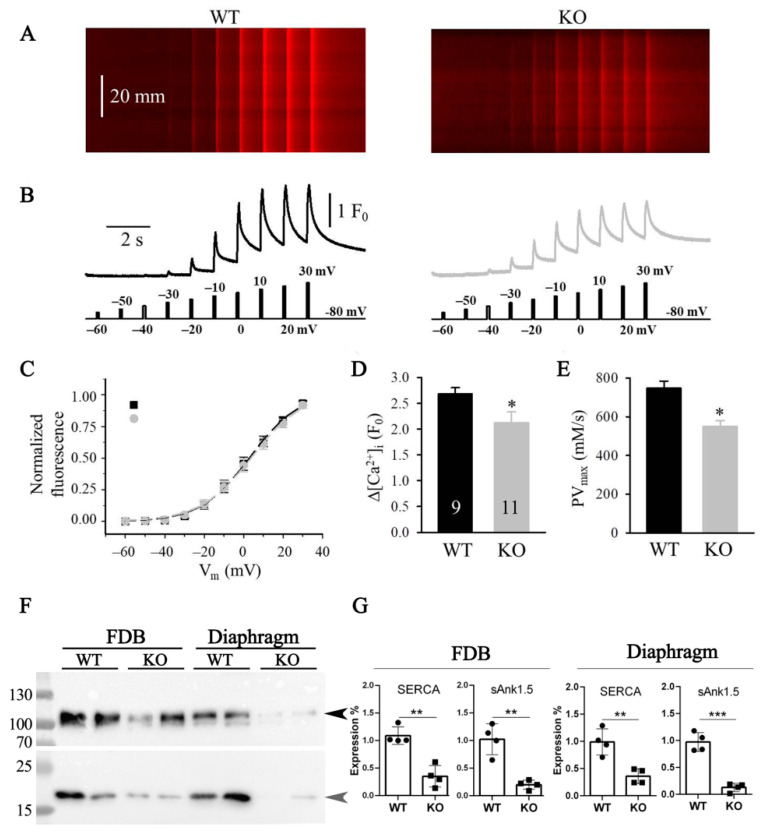
Electrically evoked Ca^2+^ transients in FDB fibers from WT and *Obscn* KO mice. (**A**) Line-scan images of changes in [Ca^2+^]_i_ elicited under whole-cell voltage clamp. Each fiber was held at −80 mV and perfused with 10 mM EGTA. The scale bar in the left panel accounts for both line scans. (**B**) Temporal profile of the changes in [Ca^2+^]_i_ calculated from the corresponding images in panel A by averaging 50 lines in the spatial domain normalized to average resting *F*_0_(*x*). The stimulation protocol contained 100 ms long progressively increasing membrane depolarizations between −60 mV and +30 mV, with +10 mV increment in every 1 s. (**C**) Voltage dependence of the changes in [Ca^2+^]_i_. Values obtained for individual fibers were first fitted by a Boltzmann function (Equation (2)) and then normalized to the obtained maximum for a given fiber, and, finally, averaged over the fibers. Continuous lines represent the best fit of the Boltzmann function to the average values with parameters of V_50_ = 3.30 and 4.36 mV, and k = 12.15 and 12.93 mV, for WT (*n* = 9) and KO (*n* = 11), respectively. (**D**) Pooled data for [Ca^2+^]_i_ and (**E**) maximal speed of SERCA pump at +30 mV depolarization from four WT and five KO mice. To assess the peak change in [Ca^2+^]_i_ in (**D**), 100 ms long single pulses were used. Numbers in columns show the number of fibers averaged. Data are reported as mean values ± SEM. * denotes significant difference from WT at *p* < 0.05. (**F**) Representative immunoblot of SERCA (dark arrow, ≈110 kDa) and sAnk1.5 (grey arrow, ≈17 kDa) on total protein lysates prepared from FDB (30 µg) and diaphragm (50 µg) muscles of two WT and two *Obscn* KO mice. (**G**) Relative densitometric analysis of immunoreactive bands of SERCA and sAnk1.5 using total protein load optical density obtained by stain-free methodology and Ponceau Red staining as normalizer. Data are presented as fold change relative to the average densitometric values of WT muscles. Bars report data obtained from FDB and diaphragm muscles from four WT and four *Obscn* KO mice as mean values ± SD (experimental replicates = 2 to 4). ** and *** denote significant difference from WT at *p* < 0.01 and *p* < 0.001, respectively.

**Figure 6 ijms-23-01319-f006:**
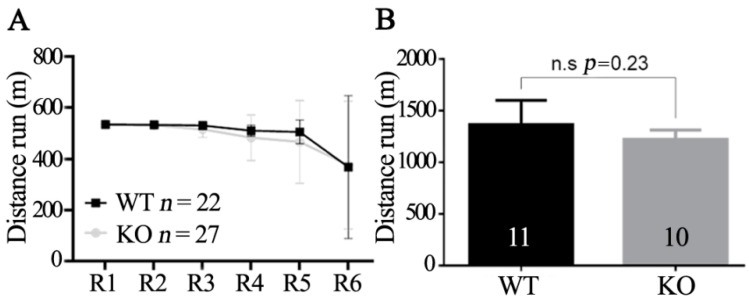
Analysis of running ability of *Obscn* KO mice. (**A**) Intense uphill running exercise. Mice performed two rounds of running (R) per day, for three consecutive days. Mean distance (m) ± SD ran in each round is reported in the graph. (**B**) Running exercise performed to exhaustion. Running session was considered concluded when mice paused running for more than 10s. Bars indicate mean distance (m) ± SD run by the two groups of mice.

**Figure 7 ijms-23-01319-f007:**
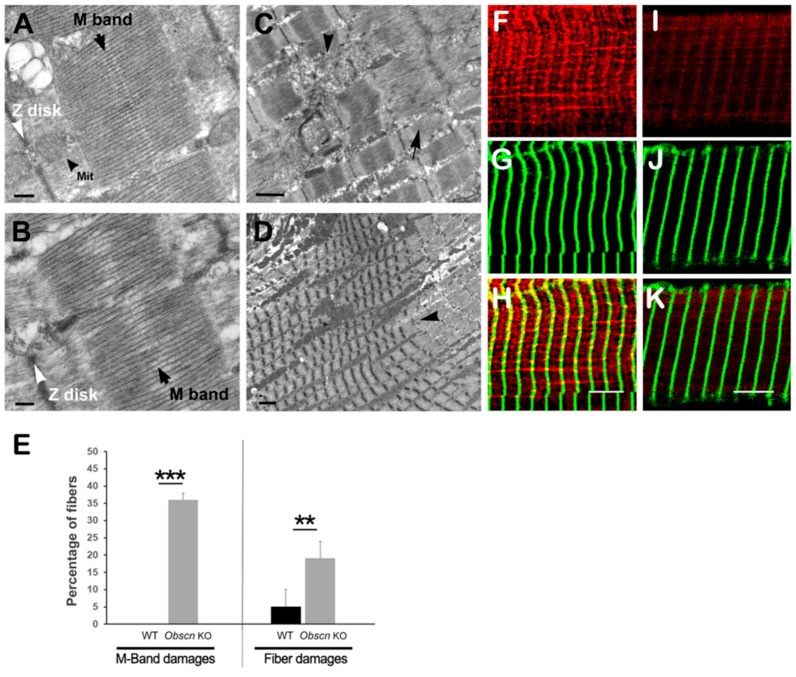
*Obscn* KO diaphragm fibers were severely damaged following intense exercise. Representative EM micrographs of WT (**A**) and *Obscn* KO (**B**–**D**) fibers following completion of intense running protocol on treadmill. Black arrows point to M-band which was properly structured in WT sarcomeres (**A**), while it was nearly collapsed (**B**) in 34% of KO diaphragm fibers (**E**). Healthy mitochondria can be easily identified in WT fibers (**A**, black arrowhead). White arrowheads point to Z-disk, which was well defined in WT sarcomere (**A**), while it was often disorganized in KO fibers (**B**). Additional fiber damages such as unstructured cores ((**C**), arrowhead), sarcomere disarray (**C**, arrow) and extensive regions with contractures (**D**, arrowhead) could be detected in 19% of *Obscn* KO fibers and only in 4% of WT fibers (**E**). Bars = 0.2 µm in A and B; 1 µm in C and D. (**F**–**K**) Immunolocalization of dystrophin. Immunostaining of EDL fibers confirmed that dystrophin (red fluorescence) localization at Z-disk, i.e., in correspondence of costameres localization, was markedly reduced in *Obscn* KO muscle fibers (**I**,**K**) compared with WT control (**F**,**H**). α-Actinin (green fluorescence) was used to decorate Z-disks (**G**,**J**). Merged fluorescence is shown in (**H**,**K**). Bars, 5 μm. ** and *** denote significant difference from WT at *p* < 0.01 and *p* < 0.001, respectively.

**Table 1 ijms-23-01319-t001:** Twitch and tetanus tension values recorded in diaphragm bundles. Mean values ± SEM.

Diaphragm	WT (4 Mice/15 Bundles)	KO (5 Mice/15 Bundles)
Twitch tension (mN/mm^2^)	52.1 ± 3.2	45.4 ± 4.3
Tetanic force 40 Hz (mN/mm^2^)	114.9 ± 11.6	104.5 ± 8.4
Tetanic force 125 Hz (mN/mm^2^)	135.3 ± 16.5	115.4 ± 9.1

**Table 2 ijms-23-01319-t002:** Time to peak (TTP) and half relaxation time (HRT) values in diaphragm bundles. Mean values ± SEM are reported. * and ** indicate significant difference from WT at *p* < 0.05 and *p* < 0.01, respectively.

Diaphragm	
WT (4 Mice/15 Bundles)	KO (5 Mice/15 Bundles)
TTP (ms) 28.3 ± 1.2	TTP (ms) 36.3 ± 1.8 **
HRT (ms) 42.9 ± 2.5	HRT (ms) 52.9 ± 3.2 *

**Table 3 ijms-23-01319-t003:** Parameters describing the voltage dependence of the depolarizing pulse-evoked Ca^2+^ transients in WT and *Obscn* KO FDB fibers. † Parameters were obtained by fitting the voltage dependence of the individual fibers using Equation (2). Mean values ± SEM. * indicates significant difference from WT (*p* < 0.05).

Boltzmann Parameters ^†^	WT (4 Mice/8 Fibers)	KO (3 Mice/11 Fibers)
max (F/F_0_)	2.70 ± 0.27	2.13 ± 0.21 *
V_50_ (mV)	2.44 ± 2.22	3.22 ± 2.43
k (mV)	11.23 ± 0.56	11.63 ± 0.66

## Data Availability

Not applicable.

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
