# Peer review of "Impaired Intracellular Ca2+ Dynamics, M-Band and Sarcomere Fragility in Skeletal Muscles of Obscurin KO Mice"

_ijms, 2022, doi:10.3390/ijms23031319_

Round 1
Reviewer 1 Report
The paper describes the effects of obscurin knockout (KO) in mice on structural and functional (physiological) properties of diaphragm and EDL muscles. This is not the first paper that deals with studying muscles in obscurin knockout mice, but, probably, the most detailed and comprehensive one. Surprisingly obscurin KO did not affect twitch or tetanus tension of diaphragm muscle although decelerated twitch development and its relaxation. Obscurin KO accelerated fatigue of bundles of diaphragm muscles caused by multiple tetanic stimulations and did not affect subsequent recovery. No effect of obscurin KO on running ability was found in vivo although fibre damage, and especially M-band damage were found in diaphragm muscle after intense exercise.
Effects of obscurin KO on calcium transients was found in the voltage-clamp experiments with FDB muscle fibres as well a reduction in the expression of SERCA and aAnk1.5.
The results are novel and important as they dissect the mechanism of obscurin involvement in the interaction of sarcomeric proteins with sarcoplasmic reticulum and cytoskeleton and its role in the maintaining of the structural integrity of muscle fibres.
Minor points
The details of the voltage clamp experiments described in 2.8.3 and legend to Fig. 5 are not sufficient for understanding the experimental protocol. A voltage plot vs time in addition to the fluorescence signal in Fig. 5B would be useful.
Papers where the effect of obscurin knockout in Drosophila on the structure of insect flight muscle (DOI 10.1242/jcs.097345 and 10.1242/jcs.170639) probably deserve citation.
Author Response
Response to Reviewer 1
We are grateful for the reviewer’ thoughtful criticisms, helpful suggestions and comments that will improve the clarity and the coherence of our manuscript.
- The details of the voltage clamp experiments described in 2.8.3 and legend to Fig. 5 are not sufficient for understanding the experimental protocol. A voltage plot vs time in addition to the fluorescence signal in Fig. 5B would be useful.
According to Reviewer’s comment, additional details concerning voltage clamp experiments have been added in paragraph 2.8.3, page 6. Figure 5 and relative legend (paragraph 3.2, page 11) have been modified according to reviewer’s suggestion.
- Papers where the effect of obscurin knockout in Drosophila on the structure of insect flight muscle (DOI 10.1242/jcs.097345 and 10.1242/jcs.170639) probably deserve citation.
The two suggested references are now correctly quoted in the manuscript as refs. 17 and 18 in the introduction section, page 2. All reference numbers have been accordingly modified.
Reviewer 2 Report
This study reports on the biomechanical properties of Obscurin KO mice skeletal muscles, emphasizing EDL and diaphragm. The authors do not observe significant changes in isometric force but report on the significantly slower kinetics of isometric contractions and delayed recovery in diaphragms from Obscn KO animals. Authors link these observations to changes in calcium handling due to the estimated reduction of SERCA activity and content in skeletal muscles of Obscn KO mice. Finally, authors report no signs of exercise intolerance in Obscn KO mice; however, they observe structural changes in diaphragms of KO animals after intense exercise. This is well written, concise report on the skeletal phenotype of Obscn KO mice.
Below are a few suggestions to improve this manuscript.
- Authors should provide images used for WB normalization.
- Provide evidence that RIPA buffer solubilizes the total pool of SERCA and sAnk1.5 and not just the RIPA-soluble fraction.
- Why was EBD uptake not assessed in the diaphragm?
- Authors should elaborate on the criteria used to identify a single muscle fiber in EM images.
- Did the authors observe sarcomeric damage in EDL muscle fibers after intense exercise?
- Authors should discuss the possibility of differences between WT and Obscn KO mice under repeated training. Would observed sarcomeric damage accumulate in Obscn KO mice and lead to exercise intolerance? Or can Obscn KO mice compensate for this damage?
- A broader discussion about the phenotype variability in genetic mice would strengthen the manuscript.
Author Response
Response to Reviewer 2
We are grateful for the reviewer’ thoughtful criticisms, helpful suggestions and comments that will improve the clarity and the coherence of our manuscript.
- Authors should provide images used for WB normalization.
The original 6 gels and blots used for WB normalization reported in Fig. 5 F and G have been now uploaded in the Supporting Information files.
We accidentally indicated the use of “ImageJ” instead of “Image Lab (Bio Rad laboratories)” to perform normalization and relative quantification of the immunoreactive bands. The use of “Image Lab (Bio Rad laboratories)” is now correctly reported in the Mat & Met section (paragraph 2.10, page 6). We apologize for the mistake.
- Provide evidence that RIPA buffer solubilizes the total pool of SERCA and sAnk1.5 and not just the RIPA-soluble fraction.
Both sAnk1.5 and SERCA are membrane proteins, which are usually efficiently solubilized by detergents and even more by RIPA Buffer. In addition, the RIPA buffer we used also contains 150 mM NaCl, which is the appropriate concentration that helps preventing protein aggregation (Bass et al., 2017; doi: 10.1111/sms.12702). Usually, RIPA insoluble fraction mainly entraps cytoskeletal, nuclear, proteasome subunits and ECM proteins. In addition, less efficient solubilization may occur with high MW proteins (over 150/200 kDa). Clearly, this does not apply to SERCA and sAnk1.5. On the other hand, the results for sAnk1.5 that we present here using RIPA buffer as lysis buffer, are superimposable to those published, using total protein extract, by Lange et al., JCS 2009, and by Blondelle et al., Commun. Biol. 2019. A tendency to decreased SERCA levels in Obscn KO mice was also observed in Blondelle et al., Commun. Biol. 2019.
- Why was EBD uptake not assessed in the diaphragm?
EBD uptake in the diaphragm was already reported in Randazzo et al., 2013. Damage to this muscle by EM has been reported in Randazzo et al., 2017 and also in the present manuscript. On these bases, we did not consider essential to repeat again the EBD uptake experiments in the diaphragm.
- Authors should elaborate on the criteria used to identify a single muscle fiber in EM images.
Determination of M-band damages or of the number of damaged fibers in WT and Obscn KO mice was performed on longitudinal oriented EM micrographs. Micrographs, all at the same magnification (18 000×) with no overlapping regions, were analyzed and individual fibers were identified as delimited by sarcolemma surrounding the sarcoplasm. Each fiber was visually scored for: 1) ultrastructural alterations and organization of the M-band with respect to the sarcomere; 2) the presence of unstructured cores and/or contracture cores, as described in Paolini et al., 2015. Quantification of the number of fibers with alterations was presented as a percentage of all analyzed fibers.
These additional details have been reported in paragraph 2.7.2 of the Mat & Met section (page 6).
- Did the authors observe sarcomeric damage in EDL muscle fibers after intense exercise?
EM analysis was performed only on diaphragm muscle that, based on our previous studies, is the most affected skeletal muscle. By histological and immunofluorescence analyses, we previously reported that EDL muscle from Obscn KO mice do not show significant overall morphological differences nor increased inflammatory cells infiltration following intense uphill running.
We only observed increased EBD uptake in EDL muscle of Obscn KO mice following in vivo electrical stimulation of hind-limb skeletal muscles that mimics intense exercise (Randazzo et al, 2017).
- Authors should discuss the possibility of differences between WT and Obscn KO mice under repeated training. Would observed sarcomeric damage accumulate in Obscn KO mice and lead to exercise intolerance? Or can Obscn KO mice compensate for this damage?
We previously reported in Randazzo et al, 2017 that, after 14 days of recovery from an intense uphill running protocol, Obscn KO diaphragm fibers recover from the initial damages and, at the histological level, no significant alterations in muscle fibers was observed. Accordingly, we exclude that sarcomeric damage may accumulate after repetition of intense exercise protocols.
On the other hand, since the running protocols used in this study are classified with the highest degree of severity by EU animal welfare legislation on use of animals in research, such experiments cannot be performed more than twice.
- A broader discussion about the phenotype variability in genetic mice would strengthen the manuscript.
The reviewer is correct in making this suggestion, but this is really an intriguing point that we tried to address but with limited success. Indeed, evidence of compensation over time in some KO mouse models has been observed and is often discussed among scientists. However, no clear answer to this question has been obtained and little is formally reported in the literature. In agreement with the reviewer’s suggestion, in the Discussion section, page 15, we added a few lines in the manuscript to underline this point, also quoting four additional references.